# Individual interactions in a multi-country implementation-focused quality of care network for maternal, newborn and child health: A social network analysis

Fidele Kanyimbu Mukinda[1]*, Nehla Djellouli[2], Kohenour Akter[3], Mithun Sarker[3], Asebe Amenu Tufa[4], Kondwani Mwandira[5], Gloria Seruwagi[6], Agnes Kyamulabi[6], Kasonde Mwaba[2], Tanya Marchant[7], Yusra R. Shawar[8,9], Mike English[10], Hilda Namakula[6], Geremew Gonfa[4], QCN Evaluation Group[¶], Tim Colbourn[2], Mary V. Kinney[1]*

1 School of Public Health, University of the Western Cape, Cape Town, South Africa, 2 Institute for Global Health, University College London, London, United Kingdom, 3 Perinatal Care Project, Diabetic Association of Bangladesh, Dhaka, Bangladesh, 4 Ethiopian Public Health Institute, Addis Ababa, Ethiopia, 5 Parent and Child Health Initiative PACHI, Lilongwe, Malawi, 6 School of Public Health, Makerere University, Kampala, Uganda, 7 Department of Disease Control, London School of Hygiene & Tropical Medicine, London, United Kingdom, 8 Department of International Health, School of Public Health, John Hopkins University, Baltimore, MD, United States of America, 9 School of Advanced International Studies, John Hopkins University, Baltimore, MD, United States of America, 10 Centre for Tropical Medicine and Global Health, University of Oxford, Oxford, United Kingdom

¶ Membership of the QCN Evaluation Group is provided in the Acknowledgments.
* mvkinney@gmail.com (MVK); fidelekmukinda@gmail.com (FKM)

**Data Availability Statement:** All data is derived from a survey conducted with stakeholders who are in settings where only one individual holds a

## Abstract

The Network for Improving Quality of Care for Maternal, Newborn and Child Health (QCN) was established to build a cross-country platform for joint-learning around quality improvement implementation approaches to reduce mortality. This paper describes and explores the structure of the QCN in four countries and at global level. Using Social Network Analysis (SNA), this cross-sectional study maps the QCN networks at global level and in four countries (Bangladesh, Ethiopia, Malawi and Uganda) and assesses the interactions among actors involved. A pre-tested closed-ended structured questionnaire was completed by 303 key actors in early 2022 following purposeful and snowballing sampling. Data were entered into an online survey tool, and exported into Microsoft Excel for data management and analysis. This study received ethical approval as part of a broader evaluation. The SNA identified 566 actors across the four countries and at global level. Bangladesh, Malawi and Uganda had multiple-hub networks signifying multiple clusters of actors reflecting facility or district networks, whereas the network in Ethiopia and at global level had more centralized networks. There were some common features across the country networks, such as low overall density of the network, engagement of actors at all levels of the system, membership of related committees identified as the primary role of actors, and interactions spanning all types (learning, action and information sharing). The most connected actors were facility level actors in all countries except Ethiopia, which had mostly national level actors. The results reveal the uniqueness and complexity of each network assessed in the evaluation.

position at various levels and within institutions. Every care has been taken to ensure the anonymity of the data in the submitted manuscript. Sharing a de-identified data set would jeopardise the conditions of informed consent. Data requests can be made to each of the Ethical Review Committees who approved this research and must be accompanied by a detailed explanation of how the data will be used in line with the ethical approvals received. Contact information for each committee are as follows: University College London Research Ethics Committee (3433/003) phone: +44 20 7679 8717, BADAS Ethical Review Committee (ref: BADAS-ERC/EC/19/00274) phone: +880 5861664150, Ethiopian Public Health Institute Institutional Review Board (ref: EPHI-IRB-240-2020) phone: +251 11 2133499, National Health Sciences Research Committee in Malawi (ref: 19/03/2264) phone: +265 789 400, and Makerere University Institutional Review Board (ref: Protocol 869) phone: +256 414 705500.

**Funding:** This work was funded by the Medical Research Council (MRC) Health Systems Research Initiative 5th call via grant MR/S013466/1 to TC at UCL Institute for Global Health, United Kingdom, YS and JS at Johns Hopkins University, United States of America, KA and AK at Diabetic Association of Bangladesh Perinatal Care Project, Bangladesh, CM at Parent and Child Health Initiative, Malawi, GS at Makerere University School of Public Health, Uganda, and ME at University of Oxford, United Kingdom; and by the Bill & Melinda Gates foundation via grant INV-007644 to TM at LSHTM, United Kingdom. The funders had no role in study design, data collection and analysis, decision to publish, or preparation of the manuscript.

**Competing interests:** The authors have declared that no competing interests exist.

They also affirm the broader qualitative evaluation assessing the nature of these networks, including composition and leadership. Gaps in communication between members of the network and limited interactions of actors between countries and with global level actors signal opportunities to strengthen QCN.

## Introduction

Low- and middle-income countries (LMICs) have been experiencing substantial improvements in the coverage of births in health facilities during the past two decades [1]; however, preventable maternal and neonatal mortality and stillbirth remains high [2, 3]. Latest published estimates show there are a combined 4.5 million deaths each year (0.29 million maternal deaths, 1.9 million stillbirths, and 2.3 million neonatal deaths) [2]. The majority of these deaths occur in sub Saharan Africa and South-East Asia. Poor quality of care during the time of birth in health facilities remains a contributor to half of these deaths globally each year [1, 4, 5].

In an effort to reduce maternal and perinatal mortality and morbidity in LMICs, 'The Network for Improving Quality of Care for Maternal, Newborn and Child Health' (QCN), established in 2017, set out to strengthen the quality of health care services [6]. Originally, the QCN comprised of nine countries (Bangladesh, Côte d'Ivoire, Ethiopia, Ghana, India, Malawi, Nigeria, Tanzania and Uganda—later joined by Sierra Leone and Kenya), which were supported by the World Health Organization (WHO), partners from the United Nations, and other stakeholder groups, such as non-governmental organizations, professional associations, and academics. The network aimed to build a cross-country platform for joint-learning around quality improvement (QI) implementation approaches and shared health outcome goals [6, 7]. As a "global network" approach, delegates from these countries formed the network, along with these global stakeholder groups, and shared learning and progress from their own nationally established QI networks through online platforms and regular meetings [7–10]. QI networks facilitate the diffusion of information between groups of people and offer an opportunity for health professionals across boundaries to learn and apply QI methods. As more countries are initiating or strengthening QI collaboratives or networks to improve healthcare services and outcomes [11, 12], such as the QCN, there is need to systematically understand their composition including the actors engaged and their level of engagement.

Global health networks are "webs of individuals and organizations" which emerge either as formal or informal entities that have the potential to evolve and influence policy and practice for health conditions or focus areas [8], in this case maternal, perinatal, and newborn health [13]. Social Network Analysis (SNA) is one method that allows for the examination of these networks by considering how individual actors interact to form social structures [14]. There are many types of networks [15, 16], two structural types include dense networks with a higher degree of interconnectedness, and less dense networks with less connections between actors displaying structural holes [17]; the latter is characteristic of most health professional organisations where groups are working in silos [18]. Other types of networks based on the various social interactions include: centralized vs decentralized networks, and integrated vs. segmented networks [19]. From this typology four other types of networks can be generated: (i) "*wheel/star*" networks (highly centralized and integrated); (ii) "*polycephalous*" networks (involving many clusters, with variable degrees of centralization); (iii) "*clique*" networks (totally decentralized and highly integrated); (iv) "*segmented, decentralized*" networks, comprising various components, made of horizontal cliques [19]. In QI collaborative networks,

there is focus on building the relationships and interactions between network actors, given that not all actors in the network are connected to each other (lower density) [20]. Collaborative relationships can, however, be hampered by actors' professional or organisational culture, differences in professional power or knowledge that can affect effectiveness and efficiency of the services provided [21]. For Provan and Julian [15], the effectiveness of a network (such as QCN) depends on the close coordination between mutually connected network subsets ('*cliques*') where services are provided by differential categories of health professionals [16].

Building QI networks is a continuous process and thus, it can go through the following four stages described by Valdis and Holley [22] that provide a typology of four distinct network structures: *(i) Scattered Fragments network*, *(ii) Single Hub-and-Spoke network*, *(iii) Multi-Hub Small-World Network*, and *(iv) Core/Periphery network*. In scattered fragment network, no connections exist or spontaneous connections are emerging between actors because no one takes the lead to build a network. In a single hub-and-spoke network, one central actor (hub) connects diverse individuals or groups based on his/her vision, social skills and links outside the network. Multiple hubs can work together in the same network (*Multi-Hub Small-World Network)*. A well-developed or mature network (core/periphery) is dense with high concentration of connections. SNA can facilitate to identify the need for shifting some connections to avoid network overload and rigidity in case of higher density [22].

This paper is part of a collection evaluating the emergence, legitimacy and effectiveness of the QCN (S1 File) [8–10, 23, 24]. As part of a broader study, the goal of applying SNA to the QCN was to explore and describe the structure of the network by mapping actors involved and examining the quality of interactions between actors. Given the complexity of the QCN, we wanted to assess the individual country networks in the four QCN countries involved in the broader study (Bangladesh, Ethiopia, Malawi and Uganda) as well as the global network. Other papers in this collection, focused on QCN emergence [8] and QCN effectiveness [10] found the conditions to be most favourable for the network in Bangladesh, followed by Ethiopia, Uganda and Malawi. We focused specifically on the following domains—information sharing, collective learning and taking collective action to improve the quality of care; these are related to the QCN strategic objectives of Learning, Action, Leadership and Accountability [6]. Our specific research questions included: 1) what is the structure of the network? 2) Who are the actors involved at different levels and what is their role in the network? 3) Who are key actors in the network (based on levels and roles)? 4) What is the nature of interactions between actors across three domains: information sharing, collective learning and reported collective action to improve the quality of care?

## Methods

### Study design

A cross-sectional survey study was conducted using SNA to assess interactions among actors involved in the QCN at global level and in four countries at national and local levels: Bangladesh, Ethiopia, Malawi and Uganda. In this paper, we focus on data from the SNA to estimate the density of the network as well as the (weighted–determined by the frequency of interaction) indegree centrality as two main measures of the network properties related to the level of connectedness and the actors' role and position within the network. Box 1 presents details about the methodology, key terms and measures for SNA [17, 25].

### Study setting

The setting of each network is different with respect to political engagement, and on-going and planned activities related to maternal, newborn and child health that could be leveraged or

## Box 1. Stakeholder network analysis overview

Using mathematical tools and specialized software packages, SNA analyses can map entities, people or events (nodes) and their relationships (paths). The method involves asking respondents (egos) to identify key members (alters) in their network in relation to a question of interest, where responses to the questions may be binary, indicating the presence of a relationship, or on a continuum, reflecting the strength of the relationship [18]. SNA systematically maps the connections across individuals to show the patterns of relationships (ties) between actors (nodes), and explores their interactions and social structures.

Key terms include:

- Node: Actors that make up the network (e.g. a single actor)

- Edges: Lines (or ties) that connect the nodes together

- Bridges: Actors that facilitate information to reach those that are isolated in the network

- Brokers: Actors that facilitate the transfer of specialized knowledge between groups

- Density: The extent to which all possible relations are actually present. It measures how the network is close to completeness or the level of connectedness in a network [26].

- Centrality: Number of connections (or ties) one node has to other nodes. If a node has many ties compared with actors, this indicates that this node has a central position in the network.

- Degree centrality (In-degree): The number of immediate contacts (alters**) an actor (ego*) has in a network. It is measured by counting the number of alters adjacent to the ego. Central connectors will have higher degree centrality, while the peripheral actor will have the lowest degree centrality.

  ○ In-degree refers to the number of edges which are coming into a node, it indicates the more popular actors as receivers of ties [26].

  ○ Weighted in-degree refers to the number of in-coming edges, weighted by the weight of each edge.

- Ego* and Alters**: Ego in SNA is the focal node, the respondent. The nodes to whom ego is directly connected to are named 'alters.

be a barrier to successful emergence, legitimacy and effectiveness of QCN [8, 10, 23]. S2 File summarizes the situations for each setting to add context to our work and explain the relevance of this study. Each country has a unique governance system for managing the health care system.

## Study population and sampling

For the purposes of this study, we considered four levels of actors within each setting. Global level actors were individuals working for an international organization, regardless of sector, as well as actors who were part of other country delegations in the QCN. National level actors

**Table 1. Characteristics of respondents.**

|  |  | Bangladesh | Ethiopia | Malawi | Uganda | Global | Total |
|---|---|---|---|---|---|---|---|
| Survey sample size (n) |  | 55 | 50 | 43 | 56 | 36 | 240 |
| Responded (Response rate %) |  | 48 (87.3) | 45 (90.0) | 85 (197.7) | 113 (201.8) | 12 (33.3) | 303 (126.3) |
| *Characteristics of respondents* |  |  |  |  |  |  |  |
| **Gender** |  | n (%) | n (%) | n (%) | n (%) | n (%) |  |
|  | Male | 28 (58.3) | 39 (86.7) | 27 (31.8) | 22 (19.5) | 3 (25.0) | 119 (39.3) |
|  | Female | 20 (41.7) | 6 (13.3) | 58 (68.2) | 91 (80.5) | 9 (75.0) | 184 (60.7) |
| **Professional background** |  |  |  |  |  |  |  |
|  | Doctors | 26 (54.2) | 11 (24.4) | 1 (1.2) | 15 (13.3) | 9 (75.0) | 62 (20.5) |
|  | Nurses | 8 (16.7) | 9 (20.0) | 74 (87.1) | 93 (82.3) | 1 (8.3) | 185 (61.1) |
|  | Other | 14 (29.2) | 25 (55.6) | 10 (11.8) | 5 (4.4) | 2 (16.7) | 56 (18.5) |
| **Level of involvement in network (n=291)** |  |  |  |  |  |  |  |
|  | Facility | 31 (64.6) | 29 (64.4) | 78 (91.8) | 101 (89.4) | - | 239 (82.1) |
|  | Sub-national | 3 (6.3) | 7 (8.9) | 6 (7.1) | 8 (7.1) | - | 24 (8.2) |
|  | National | 13 (27.1) | 9 (20.0) | 1 (1.2) | 4 (3.4) | - | 27 (9.3) |
|  | Global | 1 |  |  |  | - | 1 (0.3) |
| **Role in the network** |  |  |  |  |  |  |  |
|  | Frontline health worker | 11 (22.9) | 29 (64.4) | 80 (94.1) | 92 (81.4) |  | 212 (70.0) |
|  | Implementing partner | 13 (27.1) | 4 (8.9) | - | 2 (1.8) | 5 (41.7) | 24 (7.9) |
|  | Member of any related committee | 20 (41.7) | 10 (22.2) | 4 (4.7) | 16 (14.2) | 1 (8.3) | 51 (16.8) |
|  | Technical partner | 1(2.1) | 2 (4.4) | 1 (1.2) | 3 (2.7) | 6 (50.0) | 13 (4.3) |
|  | Other | 3 (6.3) |  |  | - |  | 3 (1.0) |

were individuals working primarily for or with national government, including Ministry of Health, donor agencies, non-governmental organizations. Sub-national level and facility actors were individuals working primarily for or with subnational or facility health management, respectively. Shawar and colleagues name and descibe the different actors engaged in more detatil [8].

Specific to the SNA analysis, 303 respondents were selected among those involved in the QCN, as identified through the evaluation [8, 10, 23], using purposeful and snowballing sampling approaches based on respondents' expertise and membership in the network. They were from national and local levels in Bangladesh (n = 48), Ethiopia (n = 45), Malawi (n = 85), and Uganda (n = 113), as well as global level actors (n = 12). Characteristics of respondents varied by network; for instance, there were less females in Ethiopia as compared to more in Malawi and Uganda (Table 1). In each country network, different units were considered to ensure a well-stratified sample including facility, sub-national and national level. Within these units, the sampling also took into consideration the primary role of actors in the network as well as their professional backgrounds. [Details on country sampling approach including snowballing which led to ~200% response rates in Malawi and Uganda].

## Data collection and analysis

Data collection and analysis was done following the steps described by Blanchet and James [27]. First, a closed-ended structured questionnaire (S3 File) was developed, pre-tested in collaboration with co-authors familiar with the local context. The first part of the questionnaire included questions on respondents' basic characteristics such as sex, current job (cadre) and their role in the network specific to each country. The second part explored the domains of interaction networks within the QCN, starting by establishing the existence of interaction between actors, to include the frequency and the quality of interaction (Box 2).

Box 2. Components of data collection tool

Domains and related questions

- Establishing interactions (Yes/No): Have you interacted with this individual on the Quality of Care Network?

- Frequency of interactions: Please indicate how often you interact with this individual on matters related to the Quality of Care Network (never = 1, annually = 2, bi-annually = 3, quarterly = 4, monthly = 5, weekly = 6, daily = 7)

- Quality of interactions (yes/no):

  ○ Collective learning: Have you undertaken some learning activities related to Quality of Care Network with this individual?

  ○ Taking actions: Have you taken forward actions related to Quality of Care Network with this individual?

  ○ Information-sharing: Have you shared information related to the Quality of Care Network with this individual?

A list of names (roster) was collated by co-authors involved in the evaluation drawing from the results of the QCN evaluation study components already completed [10]. Each survey included a list of between 20–30 names of stakeholders by global, national, and subnational/facility level who had already been identified in the broader study as part of the network. Individuals completing the survey were asked a set of questions for each stakeholder on the list (Box 2) and could add names of other individuals who they interacted with around the QCN network. For each question, a list was presented to the respondents (egos) from which they had to select with whom they interacted with (alters) [19]. Respondents were allowed to add any names not included in the list. For the 'frequency of interaction', egos had to indicate with the corresponding number, how often they interacted with each alter. For the other questions, respondents were requested to indicate with a tick the people relevant to each question. To ensure high turnout of participation in the survey, in settings where internet connectivity was available, a web-based survey was used; where internet connection was reported as a problem, a paper-based approach was followed. Specific contextual adaption to the data collection approach was done for each country (Table 2). Dissemination of the survey included face-to-face questionnaires, email, announcements during meetings, and paper-based questionnaires disseminated to key stakeholders.

Data were entered into the online survey, using the UCL-based online survey tool Opinio, by the participants who completed the online survey or country specific co-authors who entered in the data on this platform from paper-based questionnaires (AAT, MS, CN, AK, HN, LC). The data were exported into Microsoft Excel 2019 (Microsoft, USA) for data management and analysis. Two authors (FKM, MK) continuously crosschecked the data to correct inconsistencies and errors in consultation with other co-authors. The Excel matrices were saved as comma-delimited value (.csv) sheets and imported into Gephi V0.9.4 that was used for network visualisation and to generate directed sociographs that is, ties are indicated with a headed arrow. The direction of the arrow (edge) goes from the ego (arrow tail) point to the alter (arrowhead), singling one respondent gave the name of an individual who they interact with in the network. Reciprocal relationships are displayed by a double-headed arrow. The

**Table 2. Data collection process.**

| Network | Time frame of data collection | Dissemination of survey | Details on data entry into online platform |
|---|---|---|---|
| Bangladesh | January–March 2022 | Face-to-face interview using paper-based questionnaire with national level MOH stakeholders (by KA, MS)<br>Remote interviews (by phone) to key stakeholders at local level;<br>Online survey emailed individually to key implementing national partners with follow up after 1 week if no response. If there were gaps, email or phone was used to communicate and address issues.<br>Online survey shared to broader national level QCN stakeholders through a Zoom meeting<br>Local level data collected over phone by MS. | • Manual entry of data from paper-based questionnaire to online survey after most interviews completed. |
| Ethiopia | February-March 2022 | Face-to-face interview using paper-based questionnaire with stakeholders at national, subnational and facility level involved in QCN. | Manual entry of data from paper-based questionnaire to online survey after all interviews were completed. |
| Malawi | March 2022 | Online survey shared to broader national level QCN stakeholders through email or WhatsApp including MOH, donors, and implementing partners.<br>Paper-based questionnaires shared by study team; completed by participants on their own time; and collected by study team at local and facility level | Manual entry of data from paper-based questionnaire to online survey after all interviews were completed and questionnaires collected. |
| Uganda | February–April 2022 | Online survey emailed to QCN mailing lists via MOH to national level stakeholders<br>Face-to-face interview using paper-based questionnaire of frontline health workers | Manual entry of data from paper-based questionnaire to online survey during data collection period when internet connectivity allowed. |
| Global | February 28 – April 15 2022 | Online survey emailed to key global stakeholders (including individuals involved in other country QCN teams).<br>Reminder email sent after 2 weeks for those who did not complete. | All participants completed the online survey directly. |

graphs were generated by level of involvement and by the primary role of actors' engagement in the QCN. An actor in the network was represented by a coded circle (node). The size of the node relates to the number of respondents who identified the node.

## Positionality, rigour, reflexivity data validation

Two authors external to the QCN analysed the data (FKM, MK). Several meetings were held with the study team to discuss the findings after data analysis. Country data leads received a summary of the findings and three questions for reflection and interpretation for these meetings (see S4 File).

## Ethical considerations

Ethical approval was obtained from University College London Research Ethics Committee (3433/003), BADAS Ethical Review Committee (ref: BADAS-ERC/EC/19/00274), Ethiopian Public Health Institute Institutional Review Board (ref: EPHI-IRB-240-2020), National Health Sciences Research Committee in Malawi (ref: 19/03/2264) and Makerere University Institutional Review Board (ref: Protocol 869). An information sheet detailing the survey was provided to all respondents and formal written consent was obtained. All data is confidential and anonymised.

## Results

### Characteristics of respondents and composition of networks

Across the five surveys, the 303 respondents identified 566 actors (or nodes) engaged with the QCN, ranging from 89 actors in Ethiopia to 211 actors in Uganda. This represented a 56%

(A)

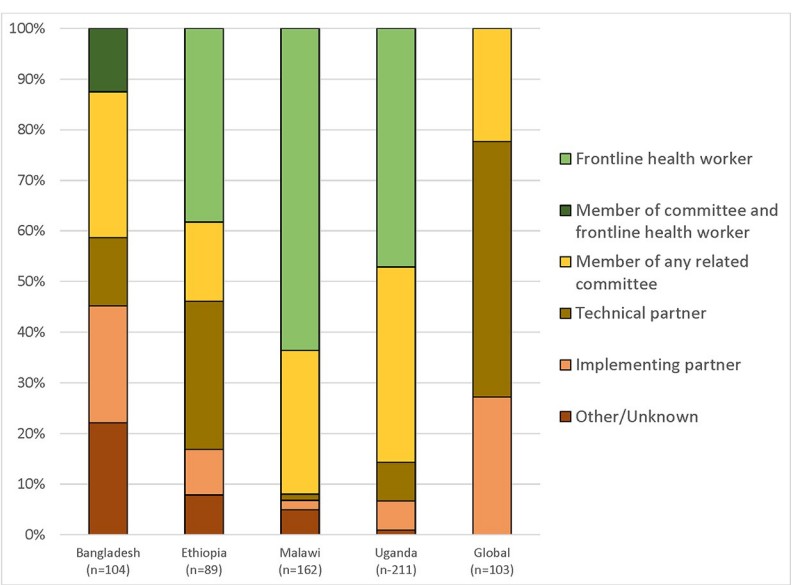

(B)

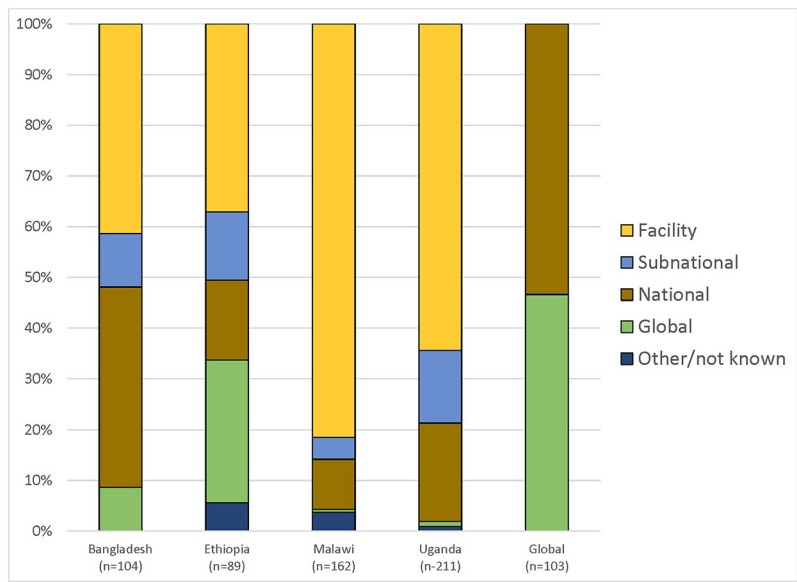

**Fig 1. Characteristics of actors identified in the network.** A: Level of actors identified in the networks. B: Primary role of actors in the network.

(303/566) network completeness, given that not all pairs of actors were connected. The actors identified in these networks worked at various levels and had a range of roles related to the network (Fig 1), including frontline health workers, member of QCN related committees, technical or implementing partner.

Respondents in Malawi and Uganda identified mostly facility level actors in their networks (81% and 64%, respectively); whereas respondents in Bangladesh and Ethiopia identified more

**Table 3. Network structures, density, central actors and interactions.**

| Network | Structure uniqueness | Network Density (% of actual connections) | Primary actors identified (based on weighted indegree) | Nature of Interactions |
|---|---|---|---|---|
| Bangladesh | Multiple-hub network including a hub of national actors interacting with each other and global level actors and multiple clusters of facility level actors connected by national and subnational actors.<br>Global actors identified as part of network linked to national level actors. | 4.6% | Top central actor: subnational level<br>Majority of top 10 actors: facility level<br>Main roles of top 10 actors: implementing partner or member of committee<br>Majority of actors: facility level | Most actors indicated that they interact across all three domains:<br>• Learning– 91%<br>• Action– 91%<br>• Information sharing– 93% |
| Ethiopia | Core/ periphery network with the majority of central actors at national level interacting with other national actors with some connections to subnational, facility and global actors. Both national and subnational actors interact with facility level actors. The structure of the network shows lots of reciprocal relationships between actors displayed by the doubled-arrowed ties.<br>Global actors identified as part of network linked to national level actors. | 5% | Top central actor: national level<br>Majority of top 10 actors: national level<br>Main roles of top 10 actors: member of committee and implementing partner<br>Majority of actors: facility level | Most actors indicated that they interact across all three domains:<br>• Learning– 87%<br>• Action– 86%<br>• Information sharing– 87% |
| Malawi | Multiple-hub network with four hubs displayed. While the majority of central actors to the network are at facility level, national actors are central connectors.<br>Global actors rarely identified by actors as part of the network. | 1.7% | Top central actor: facility level<br>Majority of top 10 actors: facility level<br>Main roles of top 10 actors: member of committee<br>Majority of actors: facility level | Most actors indicated that they interact across all three domains:<br>• Learning– 85%<br>• Action– 85%<br>• Information sharing– 84% |
| Uganda | Multiple-hub network with five hubs displayed–four facility clusters and national level cluster. The central actors involved are primarily subnational actors, although one national actor is also central.<br>Global actors rarely identified by actors as part of the network. | 1.6% | Top central actor: subnational level<br>Majority of top 10 actors: facility level<br>Main roles of top 10 actors: member of committee<br>Majority of actors: facility level | Most actors indicated that they interact across all three domains:<br>• Learning– 99%<br>• Action– 99%<br>• Information sharing– 99% |
| Global | Core/ periphery network with the majority of central actors global actors who interact mostly among themselves. One global actor is the primary connector with country level actors but other global actors also have direct interactions. | 3.1% | Top central actor: global level (World Health Organization)<br>Majority of top 10 actors: global level (World Health Organization)<br>Main roles of top 10 actors: technical partner<br>Majority of actors: global level | Most actors indicated that they interact across all three domains:<br>• Learning– 73%<br>• Action– 60%<br>• Information sharing– 80% |

national and global level actors. The global level respondents revealed about half of the actors identified were at global level (48%) and the other half among national actors across the four countries we included (15% Bangladesh, 14% Ethiopia, 12% Malawi, and 14% Uganda). Country level respondents mostly identified actors whose primary role in the network was frontline health workers (42% across all countries) or a member of a related QCN committee (30% across all countries). Ethiopia's QCN network included more technical partners (29%). For the global level, technical partner was the primary role of actors in the network (50%) with most of these actors operating at national level. Among the country actors identified in the global network, the majority were either members of committees or implementing partners (46% and 36%, respectively) (S5 File).

## Network structure, density, and key actors

The analyses revealed different types of networks (Table 3, Fig 2). The networks in Bangladesh, Malawi and Uganda were multiple-hub networks signifying multiple clusters of actors,

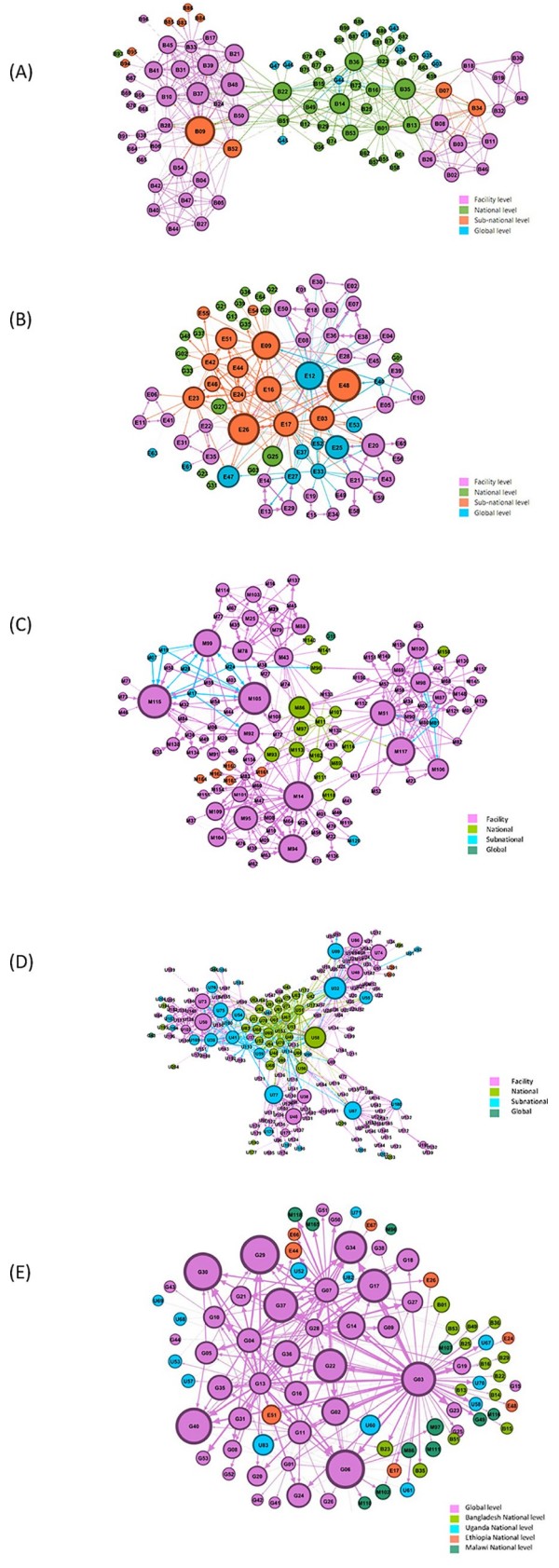

**Fig 2. Network map–interactions by frequency.** A: Bangladesh QCN Network Map. B: Ethiopia QCN Network Map. C: Malawi QCN Network Map. D: Uganda QCN Network Map. E: Global QCN Network Map.

reflected by the facility or district networks. The networks in Ethiopia and at global level were more centralized core/periphery structures with central actors at the core and other actors in the periphery. Table 3 provides further description of each network and the interactions among actors. All networks reflected the national hierarchies and the organization of the health system to some extent, with national actors often more central and serving as bridges or connectors to subnational actors, who served as connectors to facility level actors. These actors facilitate the transfer of specialised knowledge. However, national and subnational actors engaged facility actors more directly in some networks, such as Ethiopia and Malawi.

The network density was very low for all networks. Less than 10% of all potential connections were present demonstrating a low level of interactions between and across levels in the QCN (Table 3, S5 File). The networks in Bangladesh (5% density) and Ethiopia (5% density) had higher density than the other three networks (Malawi: 2% density, Uganda: 2% density, Global: 3% density; Table 3).

The central actors in each network varied (Table 3, S5 File). Actors with the highest weighted in-degree scores ranged from national level actors in Ethiopia, to subnational level actors in Bangladesh and Uganda, and facility level actors in Malawi. However, among the top 10 actors in each country network, facility level actors were dominant in all except Ethiopia, which had mostly national level actors. The primary role of these top actors across all countries was membership of a related committee. Implementing partner comprised the primary role for 23 actors (22%, Fig 1B) in the Bangladesh network. For the global network, the central actors identified worked primarily for the World Health Organization. The global actors identified by country networks were primarily global technical or implementation partners, including WHO, other UN agencies, academic organizations, and bilateral programs. Only a few respondents from the country surveys identified other country actors. For example, a Ugandan respondent added someone from the Tanzania Ministry of Health as part of their network.

Regarding the nature of interactions, the majority actors in the country networks (~90% in total) indicated that they interacted across all three domains (Table 3). There was also little variation across the different domains of interactions within each country network (learning, action, information sharing). For the global network, more actors interacted with information sharing (80%) as opposed to learning (73%) and taking forward actions (60%). The frequency of interactions varied by network (Fig 3).

Depending on the nature of the respondents, three quarters (77%) of the actors in the Bangladesh network interacted on a regular basis (daily, weekly or monthly); whereas three-quarters of the actors in the Ethiopian network interacted less regularly (quarterly, biannually, or annually). The networks in Malawi and Uganda had a more equal spread on frequency of interactions.

## Discussion

This study reveals the uniqueness and complexity of the five networks assessed for the QCN—four countries and global partners. There are some common features across the country networks, such as low overall density, engagement of actors at all levels of the system, similar roles of actors, and interactions spanning all types (learning, action and information sharing). The low network density indicates low levels of connection and interactions among QCN actors at various health system levels. Interactions were centralised around a few actors, but with little engagement and interactions among the majority of actors. Important distinctions between

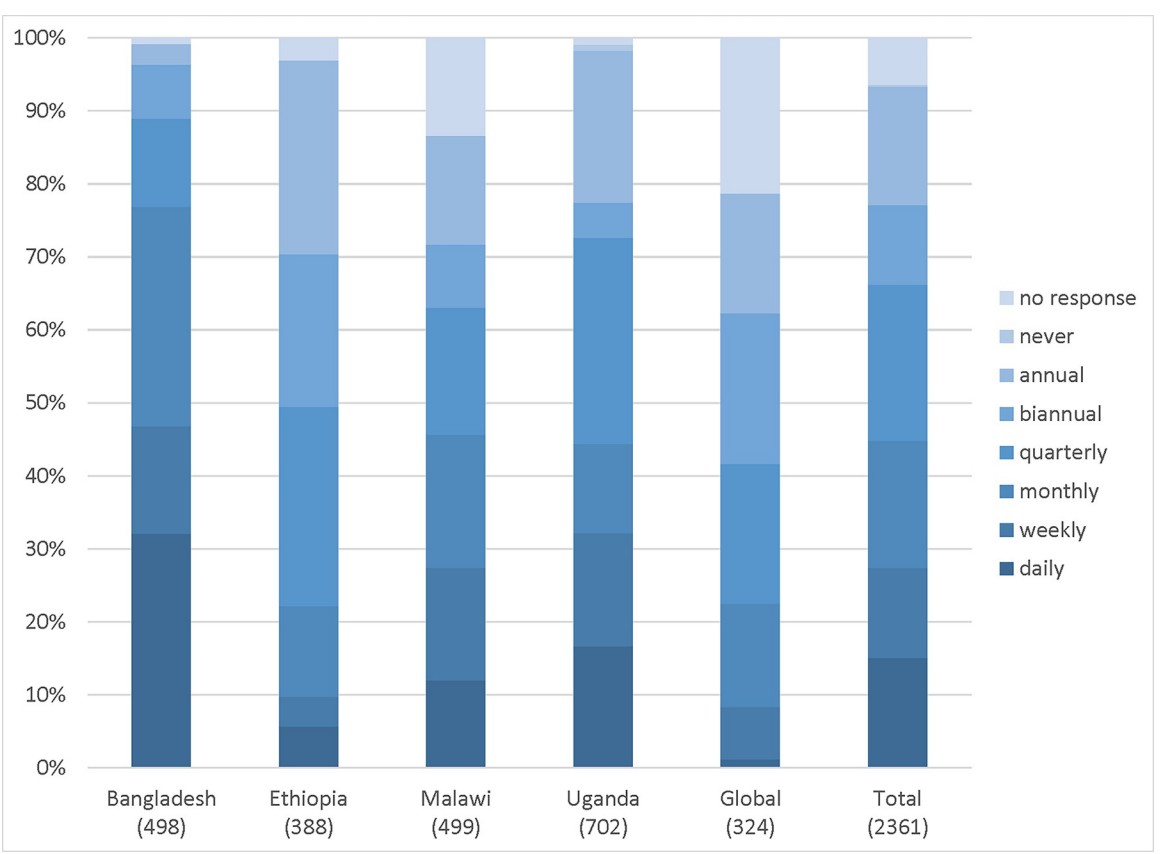

**Fig 3. Frequency of interactions by network.**

the networks include the varying frequency of interactions and structure nature, with three countries—Bangladesh, Uganda and Malawi—demonstrating multiple-hub networks.

Overall, the results display the level of interactions between network members and substantiate the qualitative findings of the QCN evaluation, especially those pertaining to political and normative interactions between stakeholder organisations [23], and the pathways through which the network emerged from the global level, through national and sub-national levels [8]. While the four country networks report to have decentralized health systems, there are some variances when it comes to implementation, as reflected by Shawar and colleagues [8], and confirmed by this study. For instance, in Ethiopia where central network position was played by the national actors, qualitative findings showed that actors from the national level were the initiators and were perceived as owners of the network; by their position, they were responsible for connecting with local levels [23]. Similarly in Malawi, central actors were identified at facility level with some connection with sub-national and national level [23]. These facility actors played a central role in the identification of quality improvement issues, development and implementation of projects. The national level was found to be less prominent in Malawi than the other three countries despite the initial strong leadership of the Quality Management Directorate in the Malawi Ministry of Health and the launch of the QCN in Lilongwe, Malawi in Feb 2017 [8]. This variation could be because the Malawi sample was mainly comprised of facility level actors, or because the actors changed over time in Malawi. The Malawi results may have differed slightly if implementing partners who were described as playing significant roles in the provision of resources for activities, technical assistance and coordination were

involved [23]. In Uganda, we also see a multi-hub network but with central actors from the national level bridging the sub-district and facility level actors facilitating, therefore, the flow of information within the network. This may reflect, not only the role national actors have had in driving the previous QI initiatives in Uganda, but also an active policy of decentralisation in the country [23].

In Bangladesh, the majority of actors identified in the SNA were from the facility and national level with national and sub-national level actors playing crucial roles to connect to facility actors, each other and global actors. This network emerged more quickly than the other countries since QCN was able to build into an existing platform demonstrated by the higher network density. Only one focal person from the national level linked with the global level [23].

At global level, the identified network through the SNA matched the observations of global meetings and interview data [14]. Much of the work of the network was coordinated by global level actors from the WHO, and technical and donor partner organisations, with national level actors in each country generally being recipients of information. National level actors also disseminated information to the global level and other partners during QCN webinars and during international meetings [9].

The SNA results came with some surprises that could be linked, in part, to the way data was collected (e.g. not surveying enough actors at national level in Malawi). Globally, participants across countries did not report interaction with each other i.e. national actors reporting interactions with other national actors at the global level, even though we know from the other studies they interacted at many regional and global meetings [10]. Even though there was no specific question in this regard, names of other national actors were included on the survey under the "global actors" category. In Ethiopia, national level respondents mentioned the influence of COVID-19 on global interactions and reported limited online interactions as well. In Bangladesh, the presence of a lay counsellor among the Top 10 with highest in-degree centrality did not align with the other qualitative research findings [8, 23].

The QCN intervention took place at multiple scales and that to some extent can explain variations in the patterns of interactions between countries. Pre-existing initiatives of implementing partners across these countries had a large influence on which facilities were chosen given prior work [23]. While this undoubtedly shaped the structure of the networks, the SNA results did not identify many implementation partners in the networks, with the exception of Bangladesh, though they are identified as key actors by the qualitative research [8, 23]. Additionally, the network patterns observed might, to some degree, reflect the political or administrative structures where there might be more decentralisation in some countries or a more centralised (command and control) form of administration in others. However, institutional and professional homophilies were reported as explanatory factors for networks formation among health professionals sharing similar interests or belonging to same organisation [28]. While respondents were asked about their professional backgrounds, there was inconsistent reporting about health cadres preventing further exploration. For example, we found many participants would identify as a specific health cadre (e.g. nurse or doctor) but their role was not clinical (e.g. Director in the Ministry of Health or Program Manager for an implementing partner).

Our study found that the primary role of top actors across all countries was 'member of a related committee' and, overall, 30% of respondents reported this as their primary role in the network. This finding aligns with the broader study which found that quality improvement committees were a core output of the QCN at facility, subnational and national level [10]. For example, in Bangladesh, there were seven committees: the Upazilla Health Complex Quality Improvement Committee (UHC-QIC), National QI Steering committee (N-QISC), National QI technical Committee (N-QITC), National Task Force Committee (N-TFC), District Quality

Improvement Committee (D-QIC), District Hospital Quality Improvement Committee (DH-QIC), and the Upazilla Quality Improvement Committee (Uz-QIC). These committees were generally perceived to have multi-disciplinary representation, to be well supported by management and aligned with government plans, though some thought coordination needed improvement [10]. In Malawi, there were four committees: The Executive and Steering Committee, the Quality of Care Coordination Team, the Quality Improvement Support Team (QIST), and the Work Improvement Team (WIT). The work of the committees in Malawi was perceived less positively overall than in Bangladesh.

The SNA provides a valuable tool to identify key actors and analyse their interactions in QI initiatives, such as the QCN; it can show the level of connectedness and the level of network fragmentation [29]. Applying the SNA to this study helps to identify which actors may be central to ensuring the QCN remains well-connected. There is a call for increased use of SNA for improvement by both the World Bank's Independent Evaluation Group and the USAID's Learning Lab [30–32]. This study is one of the few studies using SNA methodology to explore the structure and interactions of a quality of care network and is unique in its coverage of global, national and local levels of the network. Johnson and Chew [32] recently argued that "the use of SNA to improve program design, program implementation, and program evaluation and learning is quite limited" particularly in the field of international development. This study is in agreement with others that any network of care for maternal and child health can foster a multidisciplinary, multilevel teamwork, collaborative continuous learning, information sharing and problem solving [33].

There are several limitations to this study. First, not everyone involved in the network participated in the survey. The snowballing approach used for including respondents yielded not only different sample sizes, but also an over-representation of people from similar organisational structures or level of care, and this may have skewed our findings on the composition of the country networks. Conducting the SNA after the qualitative studies in each country allowed teams to identify the key actors involved and approach them for inclusion, although not all were able to participate. The SNA results only reflect information from those who responded to survey; however, the representativeness of the findings from the five networks were validated via the country teams and the qualitative data. The wide variation between the five networks also signifies the uniqueness of each context. This limits the generalisability of the findings to other countries or QI networks, and emphasises the importance of context-specific case study research.

## Recommendations

SNA can be of value to aid in planning for system improvement by identifying actors that can sustain the network beyond external support and facilitation. Interactions between countries could have happened more in QCN, and in general moving towards a denser core/periphery mature network in each country as well as global-national-local would be good for quality of care networks like QCN. Our broader work evaluating QCN found national level QCN structures were typically stronger than local structures–the periphery of the network was far weaker [10, 23]. Further work is required to strengthen the periphery of the network. This will require greater investment of time and resources at the local level and creating and strengthening bi-directional links from the centre to the periphery of the network. Furthermore, this requires increasing the frequency of interactions between and within global, national and local levels that may result in a denser/more mature network better able to facilitate improvement of quality of care.

Future study should explore how to better do SNA in complex, multilevel, multi-country collaborative networks. As alluded to by McGlashan, de la Haye [34], in such complex networks, collaborations are often centralised on a few central committee members who receive the bulk of incoming ties compared to others. Therefore, support systems should be in place to allow frequent interactions among actors within and between countries.

## Conclusion

Collaboration and interactions between cadres involved in a complex network, such as the QCN, are key ingredients for the success of such a network aiming to improve the quality of care. Our results reveal the uniqueness and complexity of each network assessed in the evaluation. They also affirm the broader qualitative evaluation assessing the nature of these networks, including composition and leadership. This study found gaps in communication between members of the network as well as limited interactions of actors between countries and with global level actors. To be effective, interactions should be strengthened between actors at all levels, particularly at the periphery that is the point of direct contact between the health system and the community receiving the services. Once established, interactive networks reduce systemic fragmentation, facilitate information sharing, learning, collective action and decision making [29]. International partners (such as WHO) can play a crucial role in strengthening individual and organisational interactions and building cohesion across levels and between countries.

## Supporting information

**S1 File. PLOS global public health QCN evaluation collection.**
(DOCX)

**S2 File. Country context.**
(DOCX)

**S3 File. Survey.**
(DOCX)

**S4 File. Data validation questions.**
(DOCX)

**S5 File. Additional results.**
(DOCX)

**S6 File. PLOS inclusivity questionnaire.**
(DOCX)

## Acknowledgments

We thank all respondents and stakeholders for their time and contributions toward making this work possible. The QCN Evaluation Group is: Nehla Djellouli, Kasonde Mwaba, Callie Daniels-Howell, Tim Colbourn (UCL Institute for Global Health, UK), Kohenour Akter, Fatama Khatun, Mithun Sarker, Abdul Kuddus, Kishwar Azad (BADAS-PCP Bangladesh), Kondwani Mwandira, Albert Dube, Gladson Monjeza, Rachel Magaleta, Zabvuta Moffolo, Charles Makwenda (Parent and Child Health Initiative, Malawi), Mary Kinney, Fidele Mukinda (independent researchers, South Africa), Mike English (Oxford University), Yusra Shawar, Will Payne, Jeremy Shiffman (Johns Hopkins University, USA), Kathy Lubowa, Agnes Kyamulabi, Hilda Namakula, Gloria Seruwagi (Makerere University, Uganda), Anene

Tesfa, Asebe Amenu, Theodros Getachew, Geremew Gonfa (Ethiopia Public Health Institute, Ethiopia), Seblewengel Lemma, Tanya Marchant (LSHTM, UK).

## Author Contributions

**Conceptualization:** Fidele Kanyimbu Mukinda, Nehla Djellouli, Kohenour Akter, Asebe Amenu Tufa, Tim Colbourn, Mary V. Kinney.

**Data curation:** Fidele Kanyimbu Mukinda, Nehla Djellouli, Kohenour Akter, Mithun Sarker, Asebe Amenu Tufa, Kondwani Mwandira, Gloria Seruwagi, Agnes Kyamulabi, Kasonde Mwaba, Hilda Namakula, Geremew Gonfa, Mary V. Kinney.

**Formal analysis:** Fidele Kanyimbu Mukinda, Mary V. Kinney.

**Funding acquisition:** Nehla Djellouli, Tim Colbourn.

**Investigation:** Kohenour Akter, Mithun Sarker, Asebe Amenu Tufa, Kondwani Mwandira, Gloria Seruwagi, Agnes Kyamulabi, Kasonde Mwaba, Hilda Namakula, Geremew Gonfa.

**Methodology:** Fidele Kanyimbu Mukinda, Mary V. Kinney.

**Project administration:** Nehla Djellouli, Tim Colbourn.

**Supervision:** Fidele Kanyimbu Mukinda, Nehla Djellouli, Tanya Marchant, Mike English, Tim Colbourn, Mary V. Kinney.

**Validation:** Fidele Kanyimbu Mukinda, Kohenour Akter, Mithun Sarker, Asebe Amenu Tufa, Kondwani Mwandira, Gloria Seruwagi, Agnes Kyamulabi, Kasonde Mwaba, Hilda Namakula, Geremew Gonfa, Mary V. Kinney.

**Visualization:** Fidele Kanyimbu Mukinda, Mary V. Kinney.

**Writing – original draft:** Fidele Kanyimbu Mukinda, Mary V. Kinney.

**Writing – review & editing:** Fidele Kanyimbu Mukinda, Nehla Djellouli, Kohenour Akter, Mithun Sarker, Asebe Amenu Tufa, Kondwani Mwandira, Gloria Seruwagi, Agnes Kyamulabi, Kasonde Mwaba, Tanya Marchant, Yusra R. Shawar, Mike English, Hilda Namakula, Geremew Gonfa, Tim Colbourn, Mary V. Kinney.

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
