## [Decision Letter · Decision Letter 0]

15 May 2023

PGPH-D-23-00370

Individual and organisational interactions, learning and information sharing in a multi-country implementation-focused quality of care network for maternal, newborn and child health: a social network analysis

Dear Dr. Kinney,

Thank you for submitting your manuscript to PLOS Global Public Health. After careful consideration, we feel that it has merit but does not fully meet PLOS Global Public Health’s publication criteria as it currently stands. Therefore, we invite you to submit a revised version of the manuscript that addresses the points raised during the review process.

The mansucript has been evaluated by 2 reviewers and their comment may be seen below.

Notably, Reviewer #1 provided a comprehensive evaluation of the manuscript and offered insightful recommendations for enhancing the clarity and reproducibility of the study's reporting.

We look forward to receiving your revised manuscript.

Kind regards,

Lucinda Shen, MSc

Staff Editor

Journal Requirements:

2. We noticed that you used "unpublished" in the manuscript. We do not allow these references, as the PLOS data access policy requires that all data be either published with the manuscript or made available in a publicly accessible database. Please amend the supplementary material to include the referenced data or remove the references.

4. We do not publish any copyright or trademark symbols that usually accompany proprietary names, eg  ©, ®, ™  (e.g. next to drug or reagent names). Please remove all instances of trademark/copyright symbols throughout the text, including ® on page 9.

Additional Editor Comments (if provided):

Reviewers' comments:

Reviewer's Responses to Questions

**Comments to the Author**

1. Does this manuscript meet PLOS Global Public Health’s publication criteria? Is the manuscript technically sound, and do the data support the conclusions? The manuscript must describe methodologically and ethically rigorous research with conclusions that are appropriately drawn based on the data presented.

Reviewer #1: No

Reviewer #2: Yes

2. Has the statistical analysis been performed appropriately and rigorously?

Reviewer #1: N/A

Reviewer #2: Yes

3. Have the authors made all data underlying the findings in their manuscript fully available (please refer to the Data Availability Statement at the start of the manuscript PDF file)?

Reviewer #1: Yes

Reviewer #2: Yes

4. Is the manuscript presented in an intelligible fashion and written in standard English?

Reviewer #1: No

Reviewer #2: Yes

5. Review Comments to the Author

Reviewer #1: The paper addresses an important issue, the quality of care for mothers and children, through learning via networks. Social network analysis is a very interesting and promising method for this, since social networks reveal the potential and improvement areas for information sharing and learning, not only to the researchers, but also to the actors involved.

Overall, the data collected is extensive, probably extra challenging to do this across borders. The methods are very well adapted to local circumstances, using different data collection techniques, both on paper and digitally. This testifies to the persistence and perseverance with the researchers and certainly is worth a compliment.

However, the paper lacks preciseness and/or explanation in several respects, which makes it difficult for the reader to understand the relevance and implications of this study. I will go into that further per section of the paper, in the attached document.

Reviewer #2: This is an important contribution to the global health community given the use of multi-country collaboratives like the QCN. The social network analysis reveals some interesting gaps in how the network is working with significant room for improvement at global level and in the 4 countries.

6. PLOS authors have the option to publish the peer review history of their article (what does this mean?). If published, this will include your full peer review and any attached files.

**Do you want your identity to be public for this peer review?** For information about this choice, including consent withdrawal, please see our Privacy Policy.

Reviewer #1: **Yes: **Annelies van der Ham

Reviewer #2: No

---

## [Decision Letter · Decision Letter 1]

31 Jul 2023

PGPH-D-23-00370R1

Individual interactions in a multi-country implementation-focused quality of care network for maternal, newborn and child health: a social network analysis

Dear Dr. Kinney,

Thank you for submitting your manuscript to PLOS Global Public Health. After careful consideration, we feel that it has merit but does not fully meet PLOS Global Public Health’s publication criteria as it currently stands. Therefore, we invite you to submit a revised version of the manuscript that addresses the points raised during the review process.

Your manuscript has been re-evaluated by one of the original reviewers, who has just one minor request (see comments below). Please could you check and correct the numbers in Table 1.

We look forward to receiving your revised manuscript.

Kind regards,

Steve Zimmerman, PhD

PLOS Staff Editor

Journal Requirements:

Additional Editor Comments (if provided):

Reviewers' comments:

Reviewer's Responses to Questions

**Comments to the Author**

1. If the authors have adequately addressed your comments raised in a previous round of review and you feel that this manuscript is now acceptable for publication, you may indicate that here to bypass the “Comments to the Author” section, enter your conflict of interest statement in the “Confidential to Editor” section, and submit your "Accept" recommendation.

Reviewer #2: All comments have been addressed

2. Does this manuscript meet PLOS Global Public Health’s publication criteria? Is the manuscript technically sound, and do the data support the conclusions? The manuscript must describe methodologically and ethically rigorous research with conclusions that are appropriately drawn based on the data presented.

Reviewer #2: Yes

3. Has the statistical analysis been performed appropriately and rigorously?

Reviewer #2: Yes

4. Have the authors made all data underlying the findings in their manuscript fully available (please refer to the Data Availability Statement at the start of the manuscript PDF file)?

Reviewer #2: Yes

5. Is the manuscript presented in an intelligible fashion and written in standard English?

Reviewer #2: Yes

6. Review Comments to the Author

Reviewer #2: A minor revision to Table 1. The number of respondents for Bangladesh is listed as 47 in row 3 whereas for the characteristics (gender, professional background, level of involvement and role), it adds up to 48. The authors should review and update for consistency.

7. PLOS authors have the option to publish the peer review history of their article (what does this mean?). If published, this will include your full peer review and any attached files.

**Do you want your identity to be public for this peer review?** For information about this choice, including consent withdrawal, please see our Privacy Policy.

Reviewer #2: No

---

## [Editor Report · Decision Letter 2]

10 Aug 2023

PGPH-D-23-00370R2

Individual interactions in a multi-country implementation-focused quality of care network for maternal, newborn and child health: a social network analysis

Dear Dr. Kinney,

Thank you for submitting your manuscript to PLOS Global Public Health. After careful consideration, we feel that it has merit but does not fully meet PLOS Global Public Health’s publication criteria as it currently stands. Therefore, we invite you to submit a revised version of the manuscript that addresses the points raised during the review process.

Thank you for correcting the number of Bangladeshi responses in Table 1 form 47 to 48. However, on page 7, line 2 (just above the table), the number of 47 is still given in the text.

Also, now the row totals does not add up correctly - I think it should be 303 rather than 302.

And the percentage for the corrected cell is now inaccurate (i.e., 48/55 is not 85.5% - it is 87.3%)

Please could you check all the numbers and percentages in the table and ensure they are accurate?

We look forward to receiving your revised manuscript.

Kind regards,

Steve Zimmerman, PhD

PLOS Staff Editor
---

## [Editor Report · Decision Letter 3]

24 Aug 2023

Individual interactions in a multi-country implementation-focused quality of care network for maternal, newborn and child health: a social network analysis

PGPH-D-23-00370R3

Dear Ms Kinney,

We are pleased to inform you that your manuscript 'Individual interactions in a multi-country implementation-focused quality of care network for maternal, newborn and child health: a social network analysis' has been provisionally accepted for publication in PLOS Global Public Health.

Best regards,

Julia Robinson

Executive Editor